# Cellular Membrane Localization of Innate Immune Checkpoint Molecule CD47 Is Regulated by Radixin in Human Pancreatic Ductal Adenocarcinoma Cells

**DOI:** 10.3390/biomedicines11041117

**Published:** 2023-04-07

**Authors:** Takuro Kobori, Yui Ito, Yuka Sawada, Yoko Urashima, Takuya Ito, Tokio Obata

**Affiliations:** 1Laboratory of Clinical Pharmaceutics, Faculty of Pharmacy, Osaka Ohtani University, Osaka 584-8540, Japan; 2Laboratory of Natural Medicines, Faculty of Pharmacy, Osaka Ohtani University, Osaka 584-8540, Japan

**Keywords:** innate immune checkpoint molecule, CD47, ezrin/radixin/moesin, pancreatic ductal adenocarcinoma, cancer immunotherapy

## Abstract

In the past decade, immune checkpoint inhibitors have exhibited potent antitumor efficacy against multiple solid malignancies but limited efficacy against pancreatic ductal adenocarcinoma (PDAC). Cluster of differentiation (CD) 47, a member of the immunoglobulin G superfamily, is overexpressed in the surface membrane of PDAC and independently correlates with a worse clinical prognosis. Furthermore, CD47 functions as a dominant macrophage checkpoint, providing a potent “do not eat me” signal to enable cancer cells to evade the innate immune system. Thus, the blockade of CD47 is a promising immunotherapeutic strategy for PDAC. In this study, we determined whether ezrin/radixin/moesin (ERM) family members, which post-translationally modulate the cellular membrane localization of numerous transmembrane proteins by crosslinking with the actin cytoskeleton, contribute to the cellular membrane localization of CD47 in KP-2 cells derived from human PDAC. Immunofluorescence analysis showed that CD47 and ezrin/radixin were highly co-localized in the plasma membrane. Interestingly, gene silencing of radixin but not ezrin dramatically decreased the cell surface expression of CD47 but had little effects on its mRNA level. Furthermore, CD47 and radixin interacted with each other, as determined by a co-immunoprecipitation assay. In conclusion, radixin regulates the cellular membrane localization of CD47 as a scaffold protein in KP-2 cells.

## 1. Introduction

Pancreatic ductal adenocarcinoma (PDAC), which accounts for approximately 90% of all pancreatic tumors, is one of the most devastating gastrointestinal malignancies with the highest mortality rate [1]. Most patients are diagnosed at an advanced stage with a metastatic state and respond poorly to current chemotherapy [2,3,4]. In the past decade, immune checkpoint blockade antibodies (Abs) targeting immune checkpoint molecules such as programmed death-1 (PD-1) and PD-ligand 1 (PD-L1) have exhibited potent antitumor efficacy against multiple solid malignancies [5,6,7]. However, immune checkpoint inhibitors have limited efficacy against PDAC [8,9,10]. Therefore, there is an urgent need to develop novel therapeutic strategies against PDAC.

Cluster of differentiation (CD) 47, a member of the immunoglobulin (Ig) G superfamily, is a transmembrane protein widely expressed in most normal and tumor tissues [11,12,13]. CD47 functions as a ligand for signal regulatory protein α (SIRPα) expressed on the cellular surface of innate immune cells including macrophages [14], and behaves as a dominant macrophage checkpoint [15,16,17], thereby providing a potent “do not eat me” signal to the macrophages that enables cancer cells to evade immunosurveillance from the innate immune system [15,16,17]. While lower expression levels of CD47 are observed in normal tissues, almost all cancer cells, including PDAC, overexpress CD47, which is independently correlated with a worse clinical prognosis in a wide variety of malignant tumors [15,16,17,18]. Thus, blocking CD47 signaling is a promising immunotherapeutic strategy for diverse cancer cell types.

The ezrin/radixin/moesin (ERM) family members are the three closely related proteins that post-translationally modulate the cellular membrane localization of numerous cancer-related transmembrane proteins by crosslinking with the actin cytoskeleton [19,20,21,22]. Our laboratory recently demonstrated that the ERM family post-translationally modulates the cellular surface localization of PD-L1, which belongs to a group of IgG superfamilies similar to CD47, by serving as a scaffold protein in several cancer cell types, including PDAC [23,24,25,26,27]. However, it is unclear whether the ERM family is involved in the cellular membrane localization of CD47 through post-translational modifications in cancer cells. In this study, we determined the role of ERM family members in the cellular membrane localization of CD47 in KP-2 cells derived from human PDAC.

## 2. Materials and Methods

### 2.1. Cell Culture

The human pancreatic tubular adenocarcinoma cell line, KP-2, established by Ikeda Y. et al. [28] (JCRB0181; JCRB Cell Bank, Ibaraki, Japan), and the human uterine cervical adenocarcinoma cell line, HeLa (EC93021013-F0; KAC, Amagasaki, Japan), were cultured according to the conventional method as previously described by Kobori et al. [24] and Tanaka et al. [25], respectively.

### 2.2. siRNA Treatment

Transfection of siRNAs was conducted as previously described by Kobori et al. [24]. Briefly, the siRNAs targeting human genes of interest and non-targeting control (NC) were transfected into KP-2 cells at a concentration of 5 nM using Lipofectamine RNAiMAX at a volume of 0.20 µL/1.0 × 10^4^ cells, followed by continuous cultivation for four days. All the reagents used for siRNA treatment were purchased from Thermo Fisher Scientific (Waltham, MA, USA).

### 2.3. RT-PCR

Total RNA extraction from KP-2 cells and real-time reverse transcription (RT)-polymerase chain reaction (PCR) were performed as previously described by Kobori et al. [24]. A list of the sequences of the gene-specific PCR primers (TaKaRa Bio, Kusatsu, Japan) used is given in Appendix A.

### 2.4. Immunoblotting

Preparation of whole cell lysates from KP-2 cells and Western blotting were carried out as previously described by Kobori et al. [24]. The source and dilution ratio of all the primary and secondary Abs used and the original immunoblots are shown in Appendix A, respectively.

### 2.5. Double Immunofluorescence Staining and Confocal Laser Scanning Microscopy

Double immunofluorescence staining and confocal laser scanning microscopy (CLSM) analysis were performed as previously described [23,24], with some modifications. The source and dilution ration of all the primary and secondary Abs used are listed in Appendix A. To quantify the colocalization ratio of CD47 with ezrin and radixin, the coefficients of Pearson’s correlation and Mander’s overlap were calculated from the two- or three-dimensional reconstructed images using the NIS-Elements Ar Analysis software (Nikon Instrument, Tokyo, Japan). Appendix A shows negative control staining in KP-2 cells using each respective secondary Ab conjugated with Alexa Fluor 488 or 594 without primary Abs.

### 2.6. Co-Immunoprecipitation

Co-immunoprecipitation assays were conducted as previously described by Kobori et al. [23,24], with some modifications. The source and dilution ratio of all the primary and secondary Abs used and the original immunoblots are shown in Appendix A, respectively.

### 2.7. Flow Cytometry

The cell surface expression levels of CD47 in KP-2 cells were measured using flow cytometry as previously described [24,25], with some modifications, using allophycocyanin (APC)-conjugated anti-CD47 Ab (323124; Bio Legend, San Diego, CA, USA) and an EC800 Flow Cytometry Analyzer and EC800 software (Sony Imaging Products and Solutions, Tokyo, Japan).

### 2.8. Analysis for Gene Expression of CD47 and Ezrin/Radixin in Patients with PDAC Using Gene Expression Profiling Interactive Analysis (GEPIA)

Gene expression of *CD47* and *ezrin*/*radixin* in tumor tissues derived from patients with PDAC and normal tissues was assessed using the GEPIA database [29]. The database is a freely available interactive web server used for analyzing the RNA-sequencing (seq) expression data of 9736 tumors and 8587 normal tissues based on data from The Cancer Genome Atlas (TCGA) [30] and the Genotype-Tissue Expression (GTEx) project [31,32].

### 2.9. Analysis for Gene-Level Correlation with Survival Rate of Patients with PDAC Using UALCAN

The relationship between the prognosis of patients with PDAC and the relative mRNA expression of *CD47* and *ezrin*/*radixin* levels in the tumor tissues was assessed with UALCAN, an interactive web source. This allows the assessment of the relationship between gene expression levels of interest and survival probability for patients with over 20,000 protein-coding genes in 33 different tumor types using RNA-seq data obtained from the TCGA project [33,34].

### 2.10. Statistical Analysis

Data are presented as the mean ± standard error of the mean (SEM) or the box-and-whisker plots with boxes indicating the median as well as the values of 25% and 75% quartiles and whiskers representing the range of values with jitter plots. Statistical comparisons between two and multiple groups were performed with a one-way analysis of variance (ANOVA) using GEPIA and ANOVA followed by Dunnett’s test using Prism version 3 (GraphPad Software, La Jolla, CA, USA), respectively. Correlation analysis was performed with Pearson’s correlation coefficient. Differences were considered statistically significant at *p* < 0.05.

## 3. Results

### 3.1. Gene and Protein Expression of CD47 and the ERM Family in KP-2 Cells

We checked the mRNA expressions of CD47 and ERM family members in 50 human pancreatic adenocarcinoma cell lines registered in the public databases of the Cancer Cell Line Encyclopedia (CCLE) and Cancer Dependency Map (DepMap) portal [35,36,37]. Database analysis indicated that the mRNA expressions of CD47, ezrin, and radixin were detected in KP-2 cells, while that of moesin was located near the limit of detection (Figure 1a), which is consistent with real-time RT-PCR data (Appendix A) and our previous observations [24]. Similarly, Western blot analysis revealed the presence of CD47, ezrin, and radixin proteins in KP-2 cells. In contrast, moesin was undetected in KP-2 cells, whereas it was detected in HeLa cells (Figure 1b), in which moesin is normally expressed at the mRNA and protein levels [25,38,39]. Because several studies on clinical human PDAC tissues and freshly isolated primary human PDAC revealed the presence of ezrin and radixin, but not moesin, at the mRNA and protein levels [40,41,42,43], we selected KP-2 cells in which the expression profiles of the ERM family are similar to those found in human PDAC tissues and freshly isolated primary human PDAC.

### 3.2. Colocalization of CD47 with Ezrin and Radixin in KP-2 Cells

The counterstaining of CD47 with F-actin, a typical plasma membrane marker, implies that in KP-2 cells, CD47 was principally localized in the cellular membrane (Appendix A). Double immunofluorescence staining showed that in KP-2 cells, CD47 was highly colocalized with ezrin (the coefficient of Pearson’s correlation and Mander’s overlap were 0.93 and 0.94, respectively) and radixin (the coefficient of Pearson’s correlation and Mander’s overlap were both 0.94) in the plasma membrane (Figure 2). On the other hand, negative control staining showed no fluorescence signal (Appendix A).

### 3.3. Knockdown Activity of Small Interfering (si) RNAs against Ezrin and Radixin in KP-2 Cells

We checked the inhibitory potency of small interfering (si) RNAs against ezrin and radixin on each target at mRNA and protein levels in KP-2 cells. Exposure of KP-2 cells to siRNAs against ezrin and radixin markedly diminished both the mRNA and protein expression levels of their respective targets (Figure 3), with little influence on cell viability (Appendix A).

### 3.4. siRNA-Mediated Knockdown of Radixin Decreased Cell Surface Expression Levels of CD47 in KP-2 Cells

We determined whether ezrin and radixin gene silencing affected the mRNA and cell surface membrane expression of CD47 in KP-2 cells. siRNA against ezrin had little influence on the mRNA and cell surface membrane expression of CD47 in KP-2 cells (Figure 4). However, radixin gene silencing significantly decreased the cell surface membrane expression of CD47 with little impact on its mRNA expression (Figure 4).

### 3.5. Molecular Interactions between CD47 and Radixin in KP-2 Cells

We investigated whether CD47 interacts with radixin in KP-2 cells using co-immunoprecipitation experiments. Radixin protein was detected in the whole-cell lysates (input) of KP-2 cells and immunoprecipitates with an anti-CD47 Ab but not with an isotype-matched control IgG Ab (Figure 5a). Similarly, reverse immunoprecipitation with an anti-radixin Ab detected the protein expression of CD47 but not with an isotype-matched control IgG Ab (Appendix A), suggesting an intrinsic protein–protein interaction between CD47 and radixin in KP-2 cells (Figure 5b).

### 3.6. Gene Expression Analysis of CD47 and Ezrin/Radixin in Clinical PDAC Tissue

We assessed *CD47* and *ezrin*/*radixin* gene expression levels in clinical tissue samples derived from PDAC tumors and normal tissues. The relative mRNA expression levels of *CD47* and *ezrin* in PDAC tumors were significantly higher than those in normal tissues. Additionally, the relative mRNA expression levels of *radixin* in PDAC tumors were moderately higher than in normal tissues (Figure 6).

### 3.7. Correlation Analysis of CD47 and Ezrin/Radixin Gene Expression Levels in Tumor Tissues with Survival Probability in Patients with PDAC

Finally, we assessed the relationship between the gene expression levels of *CD47* and *ezrin*/*radixin* in tumor tissues derived from patients with PDAC and their survival probabilities based on the TCGA database using UALCAN, a comprehensive and interactive web resource for in-depth analysis of cancer OMICS data [33,34]. The survival probability in PDAC patients with higher expression of *CD47* and *radixin* was not significant but substantially lower (*p* = 0.093 and *p* = 0.071, respectively) than in those with low/medium expression of *CD47* and *radixin*, respectively (Figure 7a–c). To further clarify the relationship between *CD47* and *ezrin*/*radixin* mRNA expressions in clinical PDAC tissues, we employed a gene correlation analysis of *CD47* with *ezrin* and *radixin* in PDAC. Gene correlation analysis revealed a significant positive correlation between *CD47* and *radixin* (Pearson’s correlation coefficient: 0.43) (Figure 7d). In contrast, there was no correlation between *CD47* and *ezrin*.

## 4. Discussion

In this study, we confirmed that KP-2 cells derived from human PDAC had sufficient expression of CD47, ezrin, and radixin at both the mRNA and protein levels, while displaying a lack of moesin (Figure 1; Appendix A). Previous studies have found that CD47 is highly expressed in clinical PDAC samples and numerous human PDAC cell lines [18,44,45,46,47]. Furthermore, several studies on clinical human PDAC tissues and freshly isolated primary human PDAC have revealed the presence of ezrin and radixin, but not moesin, at the mRNA and protein levels [40,41,42,43], implying that KP-2 cells have expression profiles similar to those of the ERM family found in patients with PDAC. Furthermore, the confocal laser scanning microscopy (CLSM) analysis provided evidence for the first time that CD47 was highly colocalized with ezrin and radixin in the surface membrane of KP-2 cells (Figure 2). Altogether, KP-2 cells have a expression profile of CD47 and ERM analogous to that found in human clinical PDAC tissues and primary human PDAC, offering an optimal experimental model for examining the involvement of the ERM family in the cellular membrane localization of CD47 in human PDAC.

CD47 is a transmembrane protein commonly overexpressed in the surface membrane of multiple cancer cells to protect them from phagocytosis upon direct interaction with SIRPα expressed on the macrophage surface membrane [48]. A growing body of evidence has demonstrated that CD47 expression is endogenously modulated by many cellular events, including transcriptional, translational, and post-translational modifications such as glycosylation and ubiquitination [48,49]. However, few studies have focused on the post-translational modification process of CD47, although it can affect its cellular membrane localization and functionality [49]. Members of the ERM family have recently emerged as key regulators of numerous cancer-related transmembrane proteins involved in cellular surface membrane localization and functional activity through post-translational modifications [19,26,50]. Importantly, we have demonstrated that the ERM family post-translationally regulate the cellular membrane localization of PD-L1, whose structure is similar to that of CD47, through molecular interactions, presumably by working as scaffold proteins in several cancer cell types, including PDAC [23,24,25,26,27,51]. Our recent study identified radixin as the predominant scaffold protein that modulates the cellular membrane localization of PD-L1 in KP-2 cells [24]. In this study, RNA interference-mediated knockdown of radixin but not ezrin significantly decreased the cell surface expression levels of CD47 without changes in its mRNA level in KP-2 cells (Figure 4). Furthermore, co-immunoprecipitation experiments revealed intrinsic molecular interactions between radixin and CD47 (Figure 5). These findings imply that in KP-2 cells, radixin may primarily contribute to the cellular membrane localization of CD47 as a predominant scaffold protein through protein–protein interactions (Figure 5). These data are further supported by the lower survival probabilities of patients with PDAC with higher expression of CD47 and radixin in clinical PDAC tissues, where a positive correlation between CD47 and radixin was observed (Figure 6 and Figure 7). Differences in the ERM proteins responsible for the cellular membrane localization of CD47 may be, at least in part, due to the divergent expression profiles of ERM depending on cancer cell type. In addition, evidence indicates that radixin plays a key role in the plasma membrane localization of certain drug transporters, including multidrug resistance protein 2 (MRP-2), possibly because radixin is predominant among ERM proteins in hepatic tissues and cells [52,53,54,55,56,57]. Hence, radixin may principally regulate the cellular membrane localization of CD47 in KP-2 cells as a dominant scaffold protein, as is the case with MRP-2 in hepatocytes. However, this issue has yet to be elucidated and should be addressed in future studies.

In summary, we observed cellular membrane co-localization and protein–protein interactions between CD47 and radixin in KP-2 cells. Furthermore, radixin may be the predominant scaffold protein responsible for the cellular membrane localization of CD47 in KP-2 cells, possibly through post-translational modifications. Thus, modulating CD47 expressions in the cellular membrane of PDAC by therapeutic agents targeting radixin such as siRNA may be an attractive immunotherapeutic strategy independent of the PD-1/PD-L1 axis. This novel macrophage checkpoint blockade therapy through its scaffold protein might offer possible therapeutic application for PDAC patients, the majority of which does not benefit from the current immune checkpoint blockade therapies.

## Figures and Tables

**Figure 1 biomedicines-11-01117-f001:**
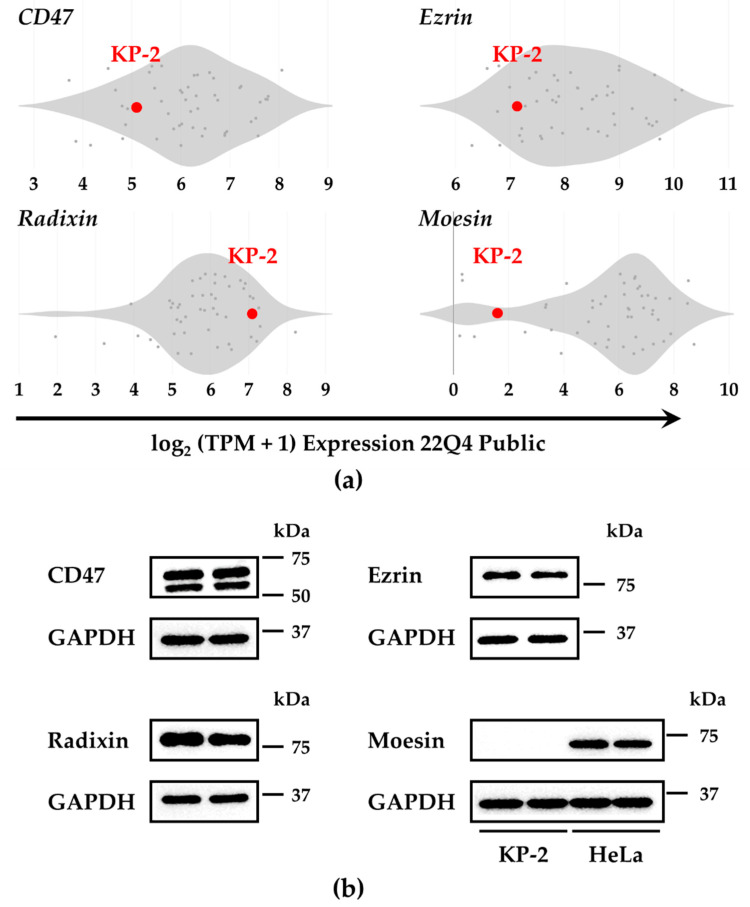
Expression pattern of CD47, ezrin, radixin, and moesin (ERM) in KP-2 cells. (**a**) Violin plots representing the median mRNA expression as log_2_ [transcripts per million (TPM) + 1] of each gene in 50 human pancreatic adenocarcinoma cell lines based on the datasets from the Cancer Dependency Map (DepMap), Broad (2022): DepMap 22Q4 Public. (**b**) Representative immunoblots for each protein in the whole-cell lysates extracted from KP-2 cells. HeLa: a positive control cell line in which moesin was endogenously expressed. GAPDH: glyceraldehyde-3-phosphate dehydrogenase, kDa: Molecular weights.

**Figure 2 biomedicines-11-01117-f002:**
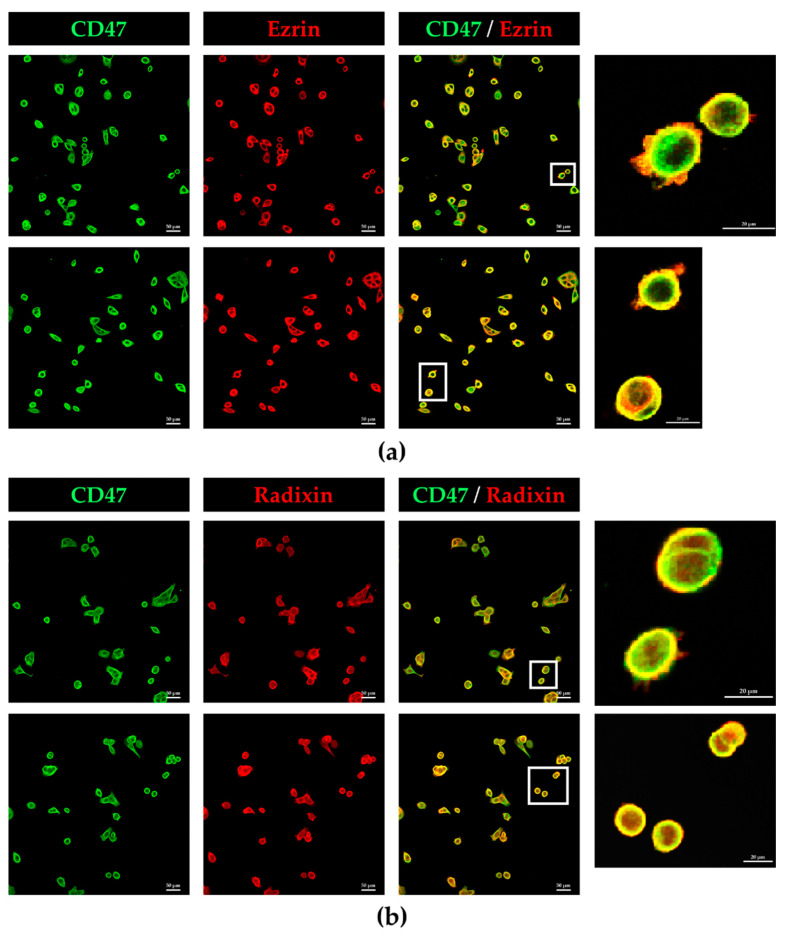
Colocalization of CD47 with ezrin and radixin in KP-2 cells. Subcellular distribution of CD47 labeled with Alexa Fluor 488 (green) and (**a**) ezrin and (**b**) radixin labeled with Alexa Fluor 594 (red) in the confocal laser scanning microscopy (CLSM) analysis. Scale bars: 50 µm. Higher magnification images in the rightmost are from the corresponding white rectangle region in the merged panels. Scale bars: 20 µm.

**Figure 3 biomedicines-11-01117-f003:**
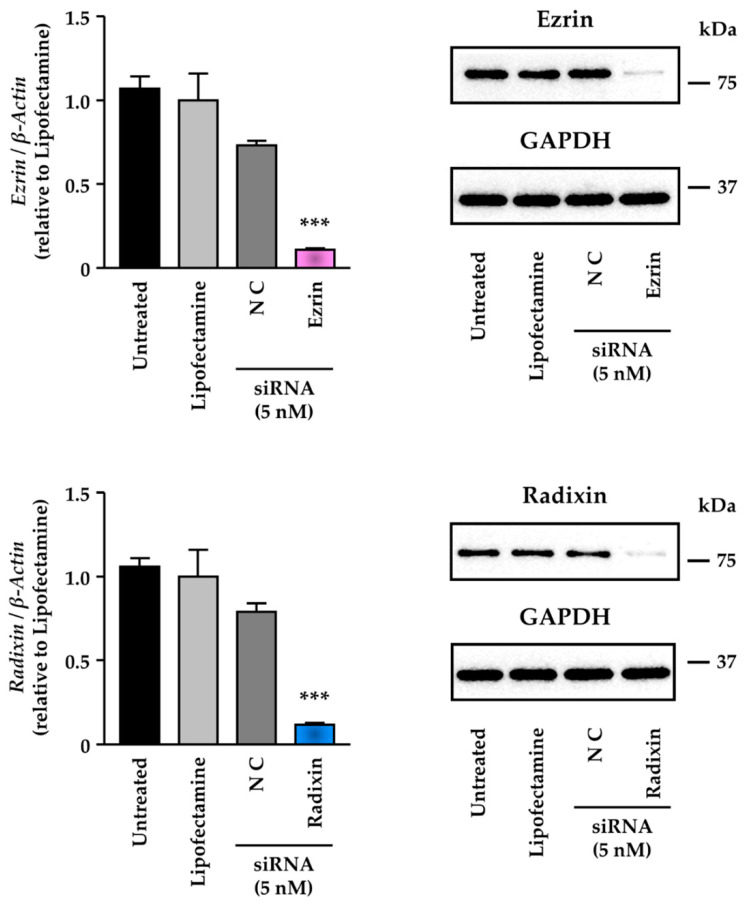
Inhibitory potencies of small interfering (si) RNAs against ezrin and adixin in KP-2 cells. (**Left**): Fold changes in the mRNA expression level of ezrin and radixin normalized to β-Actin relative to Lipofectamine alone. *n* = 3, *** *p* < 0.001 vs. Lipofectamine. (**Right**): immunoblots for ezrin and radixin. GAPDH: glyceraldehyde-3-phosphate dehydrogenases, kDa: Molecular weights.

**Figure 4 biomedicines-11-01117-f004:**
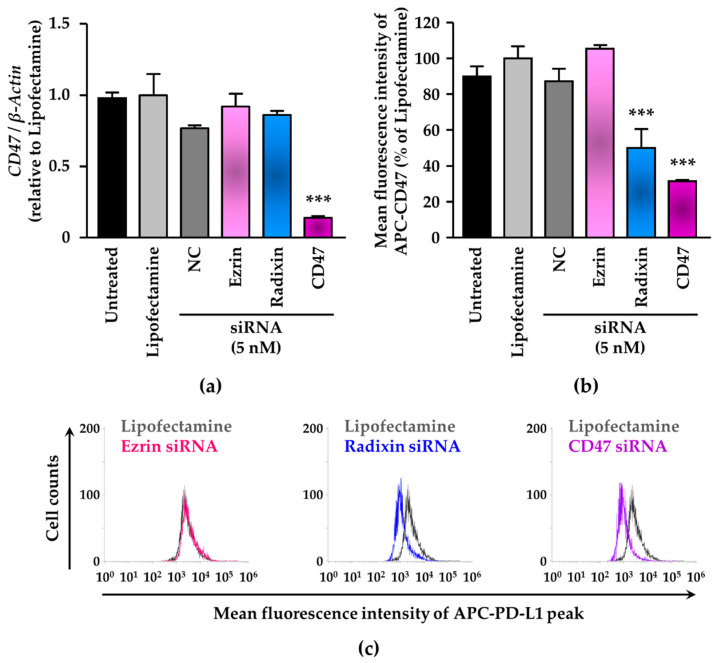
siRNA-mediated knockdown of radixin decreased cell surface expression levels of CD47 in KP-2 cells. (**a**) CD47 mRNA expression levels normalized to β-Actin in each treatment cell relative to that in Lipofectamine-treated cells; *n* = 3, *** *p* < 0.001 vs. Lipofectamine. (**b**) Mean fluorescence intensity of allophycocyanin (APC)-labeled CD47 on the cell surface membrane in all treatment groups relative to that in Lipofectamine alone; *n* = 3, *** *p* < 0.001 vs. Lipofectamine. (**c**) Flow cytometric histograms of the mean fluorescence intensity for APC-labeled CD47.

**Figure 5 biomedicines-11-01117-f005:**
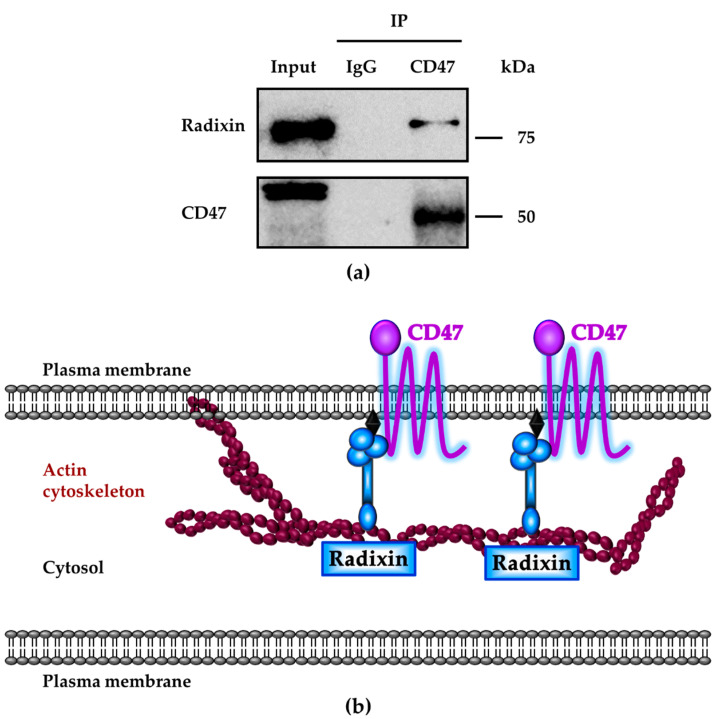
Molecular interaction between CD47 and radixin in KP-2 cells. (**a**) Representative immunoblots of CD47 and radixin in whole-cell lysates (input) and those co-immunoprecipitated with an anti-CD47 antibody (Ab) or its isotype-matched control IgG Ab. IP: Immunoprecipitation, kDa: Molecular weights. (**b**) A schematic model illustrating the molecular interaction between CD47 and radixin in KP-2 cells.

**Figure 6 biomedicines-11-01117-f006:**
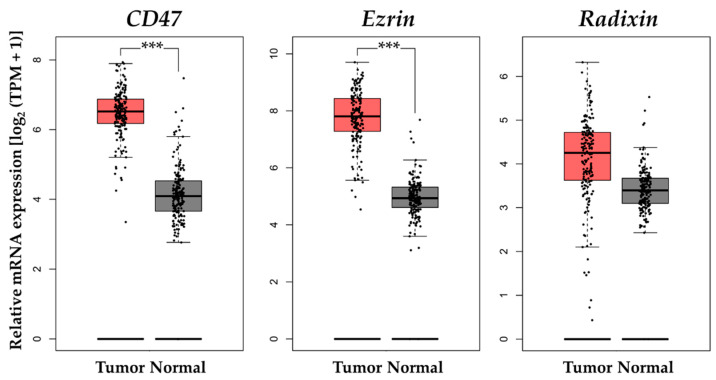
Gene expression analysis of *CD47* and *ezrin*/*radixin* in the clinical PDAC and normal tissues. Relative mRNA expression values of *CD47*, *ezrin*, and *radixin* in the tissues of patients with PDAC (*n* = 179) and normal (*n* = 177) tissues as assessed by the Gene Expression Profiling Interactive Analysis (GEPIA) database. The estimated gene expression values based on RNA-sequencing (seq) data are represented as log_2_ (TPM + 1). *** *p* < 0.001 vs. Normal.

**Figure 7 biomedicines-11-01117-f007:**
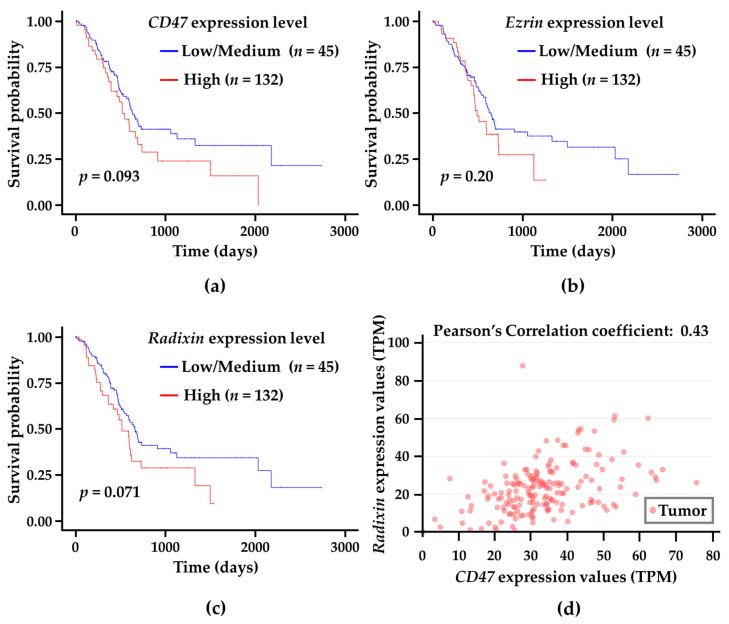
Correlation analysis for *CD47* and *ezrin*/*radixin* gene expression levels in tumor tissues with survival probability for patients with PDAC. (**a**–**c**) Survival probability curves for the patients with PDAC who had high (*n* = 132) or low/medium (*n* = 45) expression levels of *CD47*, *ezrin*, or *radixin*, respectively, as assessed by the Kaplan–Meier method using UALCAN. (**d**) Correlation analysis for the gene expression values of *CD47* with *radixin* in the clinical PDAC tissues (*n* = 179) from The Cancer Genome Atlas (TCGA) database. The estimated gene expression values based on RNA-seq data are represented as TPM.

## Data Availability

All data are contained within the article and Appendix A.

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
