# Peer review of "Cellular Membrane Localization of Innate Immune Checkpoint Molecule CD47 Is Regulated by Radixin in Human Pancreatic Ductal Adenocarcinoma Cells"

_biomedicines, 2023, doi:10.3390/biomedicines11041117_

Round 1
Reviewer 1 Report
This is a very good experimental work with good results. It would be appropriate to add to the final conclusion the possible future clinical application of the results if somehow possible.
I noted that 11 of the 62 bibliographic references are self-citations. This number is too high for one publication despite the fact that I respect and appreciate the scientific work of the authors. I think most of them should be replaced with cross-references.
Author Response
Response to Reviewer 1’s Comments
We would like to thank #Reviewer 1 for the greatest evaluation on our manuscript. We have carefully read your comments and suggestions and have made the corrections in the revised version of manuscript. Detailed responses to your comments are listed below, and we highlighted all changes with word track changes in the file labeled ‘Revised Manuscript with Track Changes’. We hope this revised manuscript would be satisfactory for publication in Biomedicines.
- It would be appropriate to add to the final conclusion the possible future clinical application of the results if somehow possible.
Reply Comments.
We would like to appreciate #Reviewer 1’s valuable suggestion. According to #Reviewer 1’s comment, we have incorporated the possible future clinical application of the present results into the Discussion section.
Discussion (Line 291–296)
Thus, modulating CD47 expressions in the cellular membrane of PDAC by therapeutic agents targeting radixin such as siRNA may be an attractive immunotherapeutic strategy independent of the PD-1/PD-L1 axis. This novel macrophage checkpoint blockade therapy through its scaffold protein might offer possible therapeutic application for PDAC patients, the majority of which does not benefit from the current immune checkpoint blockade therapies.
- I noted that 11 of the 62 bibliographic references are self-citations. This number is too high for one publication despite the fact that I respect and appreciate the scientific work of the authors. I think most of them should be replaced with cross-references.
Reply Comments.
We would like to appreciate #Reviewer 1’s polite suggestion. According to #Reviewer 1’s comment, we have removed 5 bibliographic references published from our research group as follows.
A list of references removed:
Tameishi, M.; Ishikawa, H.; Tanaka, C.; Kobori, T.; Urashima, Y.; Ito, T.; Obata, T., Ezrin Contributes to the Plasma Membrane Expression of PD-L1 in A2780 Cells. J. Clin. Med. 2022, 11, 2457.
Kobori, T.; Tanaka, C.; Tameishi, M.; Urashima, Y.; Ito, T.; Obata, T., Role of Ezrin/Radixin/Moesin in the Surface Localization of Programmed Cell Death Ligand-1 in Human Colon Adenocarcinoma LS180 Cells. Pharmaceuticals (Basel) 2021, 14, 864.
Tameishi, M.; Kobori, T.; Tanaka, C.; Urashima, Y.; Ito, T.; Obata, T., Contribution of Ezrin on the Cell Surface Plasma Membrane Localization of Programmed Cell Death Ligand-1 in Human Choriocarcinoma JEG-3 Cells. Pharmaceuticals (Basel) 2021, 14, 963.
Kobori, T.; Harada, S.; Nakamoto, K.; Tokuyama, S., Mechanisms of P-glycoprotein alteration during anticancer treatment: role in the pharmacokinetic and pharmacological effects of various substrate drugs. J. Pharmacol. Sci. 2014, 125, 242-254.
Kobori, T.; Tameishi, M.; Tanaka, C.; Urashima, Y.; Obata, T., Subcellular distribution of ezrin/radixin/moesin and their roles in the cell surface localization and transport function of P-glycoprotein in human colon adenocarcinoma LS180 cells. PLoS One 2021, 16, e0250889.
We are thankful for the time and energy you expended.
Reviewer 2 Report
The authors composed a manuscript called “Cellular Membrane Localization of Innate Immune Checkpoint Molecule CD47 is Regulated by Radixin in Human Pancreatic Ductal Adenocarcinoma Cells”. They demonstrated the association between CD47 and radixin and potentially provided a new strategy for PDAC therapy. Minor questions shall be addressed before publishing.
1. Please provide a high-magnitude of CLSM image to present the colocalization of CD47 with ezrin and radixin. Did the author quantify the colocalization rate?
2. Please justify the choice of KP-2 cells. Choosing only one cell type of PDAC is insufficient to support the potential regulation of radixin on CD47.
3. How to define high or low/medium expression levels of CD47/Ezrin/Radixin? What is the standard?
Author Response
Response to Reviewer 2’s Comments
We would like to thank #Reviewer 2 for the greatest evaluation and valuable suggestions on our manuscript. We have carefully read your comments and suggestions and have made the corrections in the revised version of manuscript. Detailed responses to your comments are listed below, and we highlighted all changes with word track changes in the file labeled ‘Revised Manuscript with Track Changes’. We hope this revised manuscript would be satisfactory for publication in Biomedicines.
- Please provide a high-magnitude of CLSM image to present the colocalization of CD47 with ezrin and radixin. Did the author quantify the colocalization rate?
Reply Comments.
According to #Reviewer 2’s comment, we have added enlarged image of each merged panel in the Figure 2. Furthermore, we have also quantified the colocalization rate of CD47 with Ezrin and Radixin in the CLSM images. Based on the additional analysis, we have incorporated these results and methods into the Figure 2 and its Legend, the sections of Materials and Methods as well as Results.
Materials and Methods (Line 88–91)
To quantify the colocalization ratio of CD47 with ezrin and radixin, the coefficients of Pearson's correlation and Mander's overlap were calculated from the two- or three-dimensional reconstructed images using the NIS-Elements Ar Analysis software (Nikon Instrument, Tokyo, Japan).
Results (Line 153–159)
CD47 was highly colocalized with ezrin (the coefficient of Pearson's correlation and Mander's overlap were 0.93 and 0.94, respectively) and radixin (the coefficient of Pearson's correlation and Mander's overlap were both 0.94) in the plasma membrane The coefficients of Pearson's correlation and Mander's overlap quantifying colocalization of CD47 with ezrin and radixin were shown on the right side of respective images. (Figure 2). On the other hands, negative control staining showed no fluorescence signal (Figure S2).
Figure 2 (Line 160)
Legend for Figure 2 (Line 163–165)
Higher magnification images in the rightmost are from the corresponding white rectangle region in the merged panels. Scale bars: 20 µm.
- Please justify the choice of KP-2 cells. Choosing only one cell type of PDAC is insufficient to support the potential regulation of radixin on CD47.
Reply Comments.
We would like to appreciate #Reviewer 2’s valuable suggestion. According to #Reviewer 1’s comment, we have incorporated the sentence to explain the reason why we have selected KP-2 cells among a number of human PDAC cell lines into the section of Materials and Methods.
Results (Line 137–141)
Because several studies on clinical human PDAC tissues and freshly isolated primary human PDAC revealed the presence of ezrin and radixin, but not moesin, at the mRNA and protein levels [44-47], we selected KP-2 cells in which the expression profiles of ERM family are similar to those found in human PDAC tissues and freshly isolated primary human PDAC.
- How to define high or low/medium expression levels of CD47/Ezrin/Radixin? What is the standard?
Reply Comments.
We would like to appreciate #Reviewer 2’s valuable comments. As #Reviewer 1 pointed out, we have rephrased the Results of Figure 1 in an objective manner.
Results (Line 130–133)
Database analysis indicated that the mRNA expressions of CD47, ezrin, and radixin were detected in KP-2 cells, while that of moesin was located near the limit of detection (Figure 1a).
We are thankful for the time and energy you expended.
Reviewer 3 Report
The manuscript by Kobori et al. "Cellular Membrane Localization of Innate Immune Checkpoint Molecule CD47 is Regulated by Radixin in Human Pancreatic Ductal Adenocarcinoma Cells" shows co-localization of CD47 and Radixin and that knockdown of radixin reduces co-precipitation of CD47 but not its mRNA. The finding is interesting. on the last figure, authors showed borderline association of CD47 and Radixin expression with survival using TCGA dataset. The authors should also examine if high CD47/high Radixin is associated with worse survival. In the result, authors stated using "numerous" cell lines of public data. Authors should always give the exact numbers of data. Fig 5a and b appear to be redundant. Overall there is not a lot of data here but the finding is interesting, so I would support the publication after revision.
Author Response
Response to Reviewer 3’s Comments
We would like to thank #Reviewer 3 for valuable suggestions on our manuscript. We have carefully read your comments and suggestions and have made the corrections in the revised version of manuscript. Detailed responses to your comments are listed below, and we highlighted all changes with word track changes in the file labeled ‘Revised Manuscript with Track Changes’. We hope this revised manuscript would be satisfactory for publication in Biomedicine.
- On the last figure, authors showed borderline association of CD47 and Radixin expression with survival using TCGA dataset. The authors should also examine if high CD47/high Radixin is associated with worse survival.
Reply Comments.
We would like to appreciate #Reviewer 3’s valuable suggestion. Unfortunately, it seems to be difficult to analyze survival plot of two genes at once in TCGA dataset using an interactive web resource UALCAN employed in this study, possibly because of the small sample size after splitting into four groups (i.e., CD47 high/radixin high, CD47 high/radixin low, CD47 low/radixin high, and CD47 low/radixin low). Meanwhile, the mRNA expression values of CD47 and radixin in the clinical PDAC tumors exhibited a positive correlation as shown in Figure 5d. Accordingly, we speculate that the survival probability in PDAC patients with higher expressions of both CD47 and radixin might tend to lower than in those with low/medium expression of both CD47 and radixin.
- In the result, authors stated using "numerous" cell lines of public data. Authors should always give the exact numbers of data.
Reply Comments.
We would like to appreciate #Reviewer 3’s valuable suggestion. As #Reviewer 3 pointed out, we have incorporated the exact number of human PDAC cells lines derived from public data base used here into the Results and the corresponding Legends for Figure 1.
Results (Line 128–130)
We checked the mRNA expression patterns of CD47 and ERM family members in 50 human pancreatic adenocarcinoma cell lines registered in the public databases of the Cancer Cell Line Encyclopedia (CCLE) and Cancer Dependency Map (DepMap) portal [35-37].
Legends for Figure 1 (Line 143–146)
Figure 1. Expression pattern of CD47, ezrin, radixin, and moesin (ERM) in KP-2 cells. (a) Violin plots representing the median mRNA expression as log2 [transcripts per million (TPM) + 1] of each gene in 50 human pancreatic adenocarcinoma cell lines based on the datasets from the Cancer Dependency Map (DepMap), Broad (2022): DepMap 22Q4 Public.
- Fig 5a and b appear to be redundant.
Reply Comments.
We would like to appreciate #Reviewer 3’s valuable suggestion. As #Reviewer 3 pointed out, Figure 5b was moved into Figure S6 in the Supplementary Materials.
Results (Line 197–199)
Similarly, reverse immunoprecipitation with an anti-radixin Ab detected the protein expression of CD47 but not with an isotype-matched control IgG Ab (Figure S6), suggesting an intrinsic protein-protein interaction between CD47 and radixin in KP-2 cells (Figure 5b).
Figure 5 (Line 200)
Legends for Figure 5 (Line 201–205)
Figure 5. Molecular interaction between CD47 and radixin in KP-2 cells. (a) Representative immunoblots of CD47 and radixin in whole-cell lysates (input) and those co-immunoprecipitated with an anti-CD47 antibody (Ab) or its isotype-matched control IgG Ab. IP: Immunoprecipitation, kDa: Molecular weights. (b) A schematic model illustrating the molecular interaction between CD47 and radixin in KP-2 cells.
Supplementary Materials (Line 301–302)
Figure S6: Reverse immunoprecipitation with an anti-radixin antibody to detect the molecular interaction between radixin and CD47 in KP-2 cells;
Supplementary Figure S6 (Line 48–53)
Reverse immunoprecipitation with an anti-radixin antibody to detect the molecular interaction between radixin and CD47 in KP-2 cells
Figure S6. Reverse immunoprecipitation with an anti-radixin antibody to detect the molecular interaction between radixin and CD47 in KP-2 cells. Representative immunoblots of CD47 and radixin in whole-cell lysates (input) and those co-immunoprecipitated with an anti-radixin antibody (Ab) or its isotype-matched control IgG Ab. IP: Immunoprecipitation, kDa: Molecular weights.
We are thankful for the time and energy you expended.
Round 2
Reviewer 3 Report
revision is sufficient, Ok to publish